# Risk Factors for Acute Renal Failure after Cardiac Catheterization Most Cited in the Literature: An Integrative Review

**DOI:** 10.3390/ijerph17103392

**Published:** 2020-05-13

**Authors:** Daniele Melo Sardinha, Alzinei Simor, Letícia Diogo de Oliveira Moura, Ana Gracinda Ignácio da Silva, Karla Valéria Batista Lima, Juliana Conceição Dias Garcez, Lidiane Assunção de Vasconcelos, Anderson Lineu Siqueira dos Santos, Luana Nepomuceno Gondin Costa Lima

**Affiliations:** 1Programa de Pós-Graduação em Epidemiologia e Vigilância em Saúde (PPGEVS), Instituto Evandro Chagas (IEC), Ananindeua 67030-000, Brazil; luanalima@iec.gov.br; 2Programa de Pós-Graduação em Enfermagem (PPGENF), Universidade do Estado do Pará (UEPA), Belém 66063-075, Brazil; alzineisimor@bol.com.br; 3Centro Universitário Metropolitano da Amazônia (UNIFAMAZ), Visconde de Souza Franco Avenue, 72, Belém-Pará 66053-000, Brazil; le_moura_@hotmail.com (L.D.d.O.M.); anagracinda08@gmail.com (A.G.I.d.S.); juliana.garcez@famaz.edu.br (J.C.D.G.); 4Programa de Pós Graduação em Enfermagem—Modalidade Mestrado Profissional em Enfermagem—UFPA, Belém 67130-600, Brazil; karlalima@iec.gov.br; 5Department of Nursing, Universidade do Estado do Pará (UEPA), Belém 66087-670, Brazil; lidianev31@gmail.com; 6Programa de Pós-Graduação em Biologia Parasitária na Amazônica (PPGBPA), Instituto Evandro Chagas (IEC) e Universidade do Estado do Pará (UEPA), Belém 66087-670, Brazil; andersonlineu@gmail.com

**Keywords:** cardiac catheterism, kidney diseases, risk factors, coronary catheterization, acute Kidney injury, acute renal failure, nephropathies

## Abstract

Acute renal failure (ARF) represents 17% of the complications of cardiac catheterization (CC), with a high death rate and longer hospitalization time. The objective of this review is to describe the most cited risk factors for acute kidney failure in the literature. It is a descriptive and exploratory Integrative Literature Review (ILR) with a qualitative approach, using articles published in the Latin American and Caribbean Health Sciences Literature (LILACS) and PubMed databases between the years of 2009 and 2019 in English, Portuguese, and Spanish, including original articles, reviews, and case studies. The search was made using the following descriptors: cardiac catheterism, kidney diseases, risk factors, coronary catheterization, acute kidney injury, acute renal failure, and nephropathies. The organization and analysis of the data was through the application of a questionnaire that was structured by the authors, and the results are presented in a table. For the final sample, 10 articles were sought. The highlighted factors were being elderly, hypertensive, and diabetic; having previous kidney disease, hypotension, heart failure, higher contrast volumes, and types; the use of non-steroidal anti-inflammatory drugs associated to other risk factors; and atrial fibrillation. Atrial fibrillation was the main finding, which has recently been documented. The identification of risk factors provides health professionals with information to plan measures to prevent ARF, minimizing complications, length of stay, and mortality.

## 1. Introduction

Cardiovascular diseases (CD) are pathologies that affect the cardiocirculatory system, resulting in high morbidity and mortality rates, representing 27.7% of the causes of death in Brazil, and becoming the main cause of death in the country [1]. CD are chronic diseases that are influenced by modifiable and non-modifiable risk factors directly related to the way of life of individuals in this century, which is characterized by sedentary lifestyles, obesity, poor diet, stress, and the presence of chronic pathologies such as systemic arterial hypertension and diabetes mellitus, which are the main cardiovascular risk factors [2].

Among chronic diseases, the following stand out: Acute Myocardial Infarction (AMI), Pulmonary Embolism (PE), Deep Venous Thrombosis (DVT), and Cerebrovascular Accident (CA) [3]. These are pathologies that represent high rates and deaths, of which AMI is the most prevalent among them, representing the main cause of deaths by chronic diseases [4]. AMI corresponds to Acute Coronary Disease, which is characterized by coronary arteries’ obstruction, induced by the presence of atheroma plaques, caused by atherosclerosis throughout life, and potentiated by the formation of a thrombus at the site of the atheroma, most often under untreated arterial hypertension. This process causes the partial or total obstruction of the lumen of the vessel, causing ischemia in the myocardium and concomitantly necrosis and death [5].

The signs of classic AMI symptoms include precordial chest pain that does not stop at rest, with or without irradiation to the jaw and upper limbs, specifically the left arm, dyspnea, cold skin, and sweating. It may also manifest with gastric pain, similar to gastritis and reflux; however, this does not occur very frequently [6]. The diagnosis is based on the presence of risk factors involved, clinical presentations, and the performance of an electrocardiogram (ECG) examination and the search for the presence of enzymes that indicate cardiac necrosis in the blood, but usually manifest after 4 h of ischemia; thus, the ECG becomes more effective for diagnosis [7].

For the AMI treatment, platelet antiaggregant drugs, thrombolytic, analgesic, and oxygen therapy are used; afterwards, it is expected that myocardial perfusion, this being the first stage of treatment. The second stage is through cardiac catheterism, which is an invasive procedure that will allow the visualization of coronary circulation, making possible to diagnose the degree of obstruction, directing it to the next conduct, angioplasty or myocardial revascularization [8].

Cardiac catheterization (CC) is a minimally invasive procedure that is performed in a hemodynamics room through the introduction of a catheter in an artery, radial, or femoral, with the objective of reaching the coronary arteries for the visualization of myocardial perfusion, in which the application of contrast media is used for the possibility of video visualization by X-ray imaging. This procedure is performed for the diagnosis of coronary diseases and for treatment such as angioplasty in the case of AMI [9].

The angioplasty procedure is a treatment for coronary artery obstructions that is performed by inserting a catheter with a balloon, making it possible to reestablish the blood flow of the affected coronary artery. So, an endovascular prosthesis (all metal) known as ‘‘Stent’’ is installed on site, which aims to keep the lumen of the vessel open and integrated, thus ensuring the perfusion of the myocardium and preventing the formation of a new obstruction in that branch [10]. Myocardial revascularization is another surgical treatment that is indicated when angioplasty is not sufficient—that is, when the obstructions are in many places. It consists of making a graft of the saphenous vein to supply the ischemia caused by the obstruction in the coronary arteries; alternatively, the mammary vein can be used used. However, everything depends on which coronary artery was affected [11].

The complications and risks offered by the cardiac catheterization procedure are important because it is an invasive method that requires anesthesia, contrast, skin, and artery lesion for catheter insertion and presents a great risk of complications such as hemorrhage at the puncture site, cardiac arrest, AMI, CA, cardiac arrhythmias, acute lung edema, hypotension, cardiac tamponade, hypersensitivity reactions, and acute renal failure (ARF) [12].

In this perspective, it is emphasized that acute renal failure complications after cardiac catheterization (contrast nephropathy) are characterized by failure in renal function, reducing glomerular filtration, causing the serum retention of nitrogen products and disturbances of the hydroelectrolytic and acid-base balance; this complication is represented by 16.5% of cases related to the cardiac catheterization procedure [13,14].

The diagnosis of contrast nephropathy is determined based on the relative elevation of 25% in serum creatinine concentration relative to basal or an absolute increase of 0.5 mg/dL in serum creatinine in the first 48 h to 72 h after the procedure. Thus, changes in the renal function are expected from the third hour after heart catheterization, although depending on the risk factors involved, such as the presence of kidney disease or kidney damage, are conditions that the ARF can occur at any time [15].

Based on this problem, the following research question emerged: What are the most cited risk factors for acute kidney failure after cardiac catheterization in the literature? The objective of this research is to describe the most cited risk factors for acute kidney failure in the literature.

## 2. Materials and Methods 

This Integrative Literature Review (ILR) features exploratory research with a quantitative approach. ILR is a method that allows the gathering of studies already published, with the objective of synthesizing evidence on a subject; it is widely used in the health sciences in order to search for methods for health care and determine innovations, thus applying evidence-based services, ensuring quality, and promoting patient safety. It has six steps that must be followed respectively: Elaboration of the research question; Inclusion and exclusion criteria; Definition of sampling; Evaluation of included studies; Interpretation of results; and Presentation of the ILR synthesis [16].

The Patient, Intervention, Comparison, Outcome (PICO) strategy is widely used in evidence-based practice, and it was used to elaborate the research question, in which it proposes that problems identified in clinical practice, research, and teaching be organized from four elements: Patient, Intervention, Comparison, Outcome (PICO). Construction from these elements provides greater scope for the resolution of the problem addressed [17].

Therefore, the research question and criteria were as follows: What are the risk factors for acute kidney failure after cardiac catheterization? Patient: Individuals undergoing cardiac catheterization >18 years/Intervention: Risk factors for acute renal failure/Comparison: Relate the progression to renal failure with the cardiac catheterization procedure/Outcome: Describe the risk factors for acute renal failure after cardiac catheterization.

Elected databases for the search: Latin American and Caribbean Health Sciences Literature (LILACS) and PubMed. We included original articles, reviews, and case studies in English, Portuguese, and Spanish that were published from 2009 to 2019. Criteria for exclusion: experience reports, dissertations, theses, guidelines, and manuals.

For the database search, the following health science descriptors were listed: cardiac catheterization; kidney diseases; risk factors; coronary catheterization; acute kidney injury; acute renal failure; nephropathies. Two search strategies have been performed, see Table 1.

After searching and defining the sample, the articles that answered the research question were read in depth. In order to organize and collect the data, a questionnaire prepared by the authors was used, consisting of the items: article number, title, authors, year, database, methodology, and risk factors for ARF after cardiac catheterization.

After data collection, with the help of the questionnaire, the information was organized in a table, thus making it possible to describe the articles’ profile and highlight the risk factors for ARF after cardiac catheterization evidenced in the included studies, synthesizing these results and presenting them for further discussion.

## 3. Results

In the search using the descriptors through strategy #1, the LILACS database resulted in 41 articles; after the inclusion criteria and reading, 4 articles were selected. In PubMed, the search resulted in 551 articles, but the majority did not specifically address the subject, so after the inclusion criteria listed, only 2 were included in this search. Through strategy #2, in LILACS, the search resulted in 1 article that was excluded. In PubMed, the search resulted in 298 articles that did not specifically address the risk factors that were analyzed and were thus excluded, or the results showed no clarity. Thus, 4 more articles were included. See the flowchart (Figure 1) of the search, which was based on the recommendations of the Preferred Reporting Items for Systematic Reviews and Meta-Analyses (PRISMA) protocol [18].

Thus, the sample was defined as 10 articles; then, the data from the questionnaires were included in the table, comprising the article number, year, database, authors, methodology, and risk factors for ARF after cardiac catheterization. For the inclusion of risk factors, after intense reading of the results of the articles included in the sample, each risk factor was highlighted and included in the table. The following Table 2 shows the information extracted from the sample of the questionnaire applied.

Regarding the year of publication, 3 articles were published in 2011, 1 article was published in 2012, 1 article was published in 2014, 1 article was published in 2010, 1 article was published in 2017, and 2 articles were published in 2018. Regarding methodology, (9) articles presented a field study in which groups of patients who underwent cardiac catheterization were analyzed, and (1) article was a systematic meta-analysis review on the evidence of atrial fibrillation as a risk factor for ARF after cardiac catheterization.

For the risk factors evidenced, see Table 3 below.

## 4. Discussion

It is known that in order to perform cardiac catheterism, iodinated contrasts are necessary to visualize the coronary arteries with the aim of diagnosis or treatment [29]. 

Studies show that the use of contrast has the potential risk of causing ARF, in which 11% of the cases of ARF come from the use of contrast for imaging exams, catheterization, and surgeries. This type of complication has a high rate of other hemodynamic complications, as well as as increase of the length of hospital stay and the mortality of affected patients. However, there are risk factors that increase the chances of developing ARF after the use of contrast [30].

The guideline for diagnostic and therapeutic tests in hemodynamics of the Brazilian Society of Cardiology (SBC) emphasizes that nowadays, organic iodine-based contrasts are used, which are classified as having high osmolarity/ ionic contrasts: variable osmolarity, and have 5–8 × plasma osmolarity. Its ph is presented as between 6.0 and 7.0, while it has a sodium concentration that is 1–7 × higher than the plasmatic one and it has additives that chelate the ionic calcium. They have expressive antithrombotic properties. Low osmolarity/ionic contrasts: with a decrease in osmolarity above 60%, thus having a decrease in the effects related to hypertonicity. Moreover, they do not chelate the ionic calcium and preserve the antithrombotic characteristics. Non-ionic have a reduced osmolarity over 50%. They do not chelate calcium ionic and present decreased antithrombotic characteristics. Isosmolar have an osmolarity similar to plasmatic, with a significant reduction in the effects related to hyperosmolarity [31].

Regarding contrast-related nephrotoxicity, iodates may cause ARF, which is characterized by a temporary increase in renal function tests (peak in 3–5 days), requiring dialysis treatment sometimes, as well as the evolution to chronic renal failure (CRF). In patients at higher risk, the literature points out the benefit of using non-ionic low osmolarity contrasts in the prevention of contrast nephropathy as well as the use of isomolar non-ionic contrasts for prevention [32].

In this way, the identification of the main risk factors favors measures for nephrotoxicity related to contrast prevention, thus highlighting the factors evidenced in this review.

It was evidenced in this study that the elderly, specifically over 75 years of age, are a group at risk for ARF after CC. These findings corroborated the results of Singh et al. [33], who evaluated the causes of ARF in 100 elderly in a Department of Nephrology, Institute of Medical Sciences, BHU, Varanasi, UP, India, from 2014 to 2015, and found that the second cause of ARF was the use of contrasts for catheterization or imaging studies. It also showed mortality in 45% of cases, which was related to the elderly already having decreased glomerular filtration, influenced by the aging process, and in most cases, the presence of chronic diseases, resulting in the need for continuous pharmacological therapies and contributing to renal nephrotoxicity.

So, it is highlighted that being elderly is an important risk factor to be considered for the development of ARF after CC, also showing that the afflictions in almost half of those affected evolve to death, thus requiring preventive measures to be taken in order to minimize cases and mortality.

Another important risk factor reported in most articles was the presence of diabetes mellitus (DM) and systemic arterial hypertension (SAH). Going against the results of Antunes et al. [34], who evaluated 54 hospitalized patients who underwent cardiac catheterization, in which the prevalence of ARF was 24.1% in 13 cases, precisely in patients with DM and SAH, being less prevalent in patients without DM and SAH. It was also noted that ARF in most patients was irreversible.

SAH is a risk factor for chronic renal insufficiency, as it causes microvascular pressure on nephrons, which are injured over time, causing hypertensive nephroesclerosis. Its presence is related to the majority of hypertensive patients already having a kidney injury that is not yet symptomatically evident, and they are potentiated with the use of contrast, which is a nephrotoxic drug that will easily injure kidneys that already have some function impairment [35].

Regarding the presence of DM, it is highlighted that it is the main cause of ARF, because persistent hyperglycemia causes damage to nephrons, leading to a diabetic nephropathy condition and reverberating on renal function failure; thus, the use of contrast presents a high risk of evolving to ARF and even death. Patients with DM require attention when performing procedures with contrast, needing evaluation of the renal function in order to minimize severe renal damage [36].

Another highlighted risk factor was heart failure (HF), in which Selistre et al. [37] also evidenced as a risk factor in his study with 400 patients who used contrast for diagnostic tests in a hospital in Rio Grande do Sul, Brazil. Santos et al. [38] justified that HF has the potential to cause low renal perfusion due to the low left ventricle ejection fraction, causing the risk of kidney ischemia with the use of contrasts. In the same study, the author highlighted that hypotension has a similar potential, because it will cause low renal perfusion and it is also a risk factor for ARF after the use of contrast, corroborating with the results of this study, which also showed hypotension as a risk factor in one article of the sample.

This way, HF and hypotensive patients have a high risk of nephrotoxicity by contrast, because renal perfusion is already reduced by the HF pathology, thus requiring attention in patients with HF when performing CC, while for hypotensive patients, the stabilization of blood pressure levels is recommended for performing CC.

The continuous use of drugs such as non-steroidal anti-inflammatory drugs was also highlighted that as a risk factor for ARF with the use of contrast. This result was contrary to the study of Diogo et al. [39], which analyzed 236 patients who underwent cardiac catheterization, assessing the renal function after 72 h of the procedure; in this study, 42% of the patients used NSAIDs, but the prevalence of ARF was 10.37%, with no association between the use of NSAIDs and risk of ARF with the use of contrast. In this same study, it was identified that previous chronic kidney disease was more relevant as a risk factor for ARF, according to the results of this study, which also identified previous renal disease as a risk factor for ARF in CC.

In this context, Lucas et al. [40] conducted a review study on the use of NSAIDs as a factor of kidney damage, and it was shown that for patients without the presence of other risk factors for kidney disease such as SAH, DM, and the elderly, and who do not take abusive doses, there is no significant damage of renal function loss, but in the presence of any factor and risk for kidney disease, there is a potential risk of kidney injury and later ARF, as for example in the use of a nephrotoxic drug such as contrast.

The research indicates that the use of NSAIDs in the recommended doses does not result in kidney damage with risks of ARF, but the use of NSAIDs related to other risk factors increases the risk of ARF. It also highlighted that the high doses are even worse, resulting in greater damage in nephrons. The presence of previous renal disease is also a risk factor, since the individual has already some loss of function or alteration, so the use of contrast more easily harms the glomerular filtration functions.

Regarding the volume of contrast used in the procedure, it was shown that large volumes, resulting from longer procedure times, become a risk factor for ARF. Antunes et al. [34] evaluated 54 patients who received the average volume of 234.8 mL of contrast, with 209.7 mL in diagnostic procedures and 345.5 mL in therapeutic. There were 33.4% of cases of ARF, which were related to the majority of patients who received higher doses of contrast volume, being explained by the longer time of cardiac catheterization and requiring the use of more contrast. He also highlighted that doses greater than 200 mL increase the risks of ARF.

Only one of the articles included in this study pointed to atrial fibrillation (AF) as a risk factor for ARF after CC, which was evidenced through a systematic review of the literature with meta-analysis.

It was explained in this study that AF could lead to hemodynamic disorders due to rapid ventricular rates, irregular heartbeat, the loss of atrioventricular synchrony, and atrial contraction, all of which resulted in insufficient cardiac output, leading to the risk of ARF, despite the normal left ventricle ejection fraction. It also showed that patients with AF load tend to have a greater expression of angiotensin-converting enzyme levels, as well as more angiotensin II receptor levels, which implies that there is more activity of the angiotensin–aldosterone renin cascade. This results in more damage to the kidney. He also related that AF is associated with a higher inflammatory state, as reflected by fibrotic changes in the myocardium and kidney. Thus, it may lead to renal impairment due to high susceptibility to injury. It was also highlighted that micro renal emboli may result from AF and lead to ARF [24].

A study conducted by Sedhai et al. [41] with 513 hospitalized patients who underwent cardiac catheterization estimated the prevalence of ARF in 3.7% of patients and suggested that these cases were responsible for long hospitalization and higher mortality; it was also noted that ARF was more prevalent in patients with pre-existing AF, congestive heart failure (CHF), and CKD. When tested by univariate analysis, the incidence of ARF was 13.8% in the group with AF (<0.002), compared with 2.3%, 1.9%, and 2.4% in the absence of pre-existing AF, CHF, and CKD, respectively. In additional tests using a multivariate logistic regression model using AF, CHF, and CKD as independent variables, the development of ARF was strongly associated with AF with an odds ratio of 4.11, 95% CI: 1.40–12.07, *p* = 0.01. These results showed that AF is a risk factor for ARF after CC, corroborating the results of this study.

This factor of AF is highlighted: little is discussed and known in the research conducted in Brazil, which is the country of origin of the authors. Thus, this review is of great relevance, allowing a better stratification of risks of patients undergoing cardiac catheterization.

## 5. Conclusions

Through this review, it was possible to describe the main risk factors cited in the literature related to the development of acute kidney failure as a complication after cardiac catheterization. The factors highlighted were elderly, hypertensive and diabetic, previous kidney disease, hypotension, heart failure, higher contrast volumes and types, the use of NSAIDs associated with other risk factors, and atrial fibrillation.

As atrial fibrillation is a risk factor that has been recently documented in the literature, the review studies allow evidence-based health service practice, as it provides new studies that will subsidize the practice with more quality. Describing the risk factors for ARF after cardiac catheterization allows a better stratification of the risk factors in patients and the prevention of renal damage, since it has been shown that the use of ARF increases the period of hospitalization and is directly related to a higher mortality rate in affected patients.

Thus, this study provides information based on evidence for professionals and academics, as well as being a way to promote population health and the practice of patient safety, since knowing the risk factors makes it possible to work on prevention, reducing public spending and impacting on the health and quality of life of patients undergoing cardiac catheterization.

## Figures and Tables

**Figure 1 ijerph-17-03392-f001:**
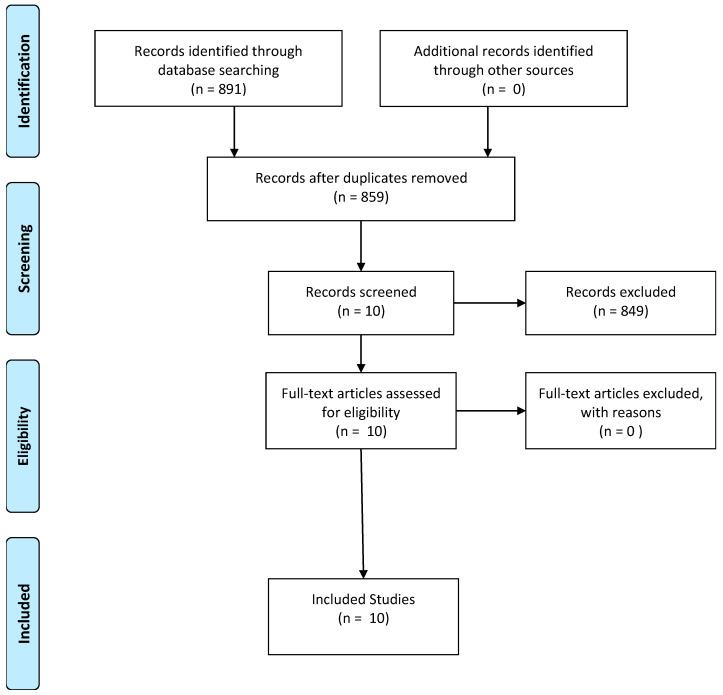
Study selection flowchart according to Preferred Reporting Items for Systematic Reviews and Meta-Analyses (PRISMA 2009) adapted. Source: Authors’ research.

**Table 1 ijerph-17-03392-t001:** Search strategies in electronic database.

Strategy	Keywords
#1	Cardiac Catheterization; Kidney Diseases; Risk Factors
#2	Coronary Catheterization; Acute Kidney Injury; Acute Renal Failure; Nephropathies.

**Table 2 ijerph-17-03392-t002:** Summary of the results from the questionnaire applied to the sample articles in the Latin American and Caribbean Health Sciences Literature (LILACS) and PubMed databases. ARF: acute renal failure.

No. Title Year/Base	Authors	Methodology	Participant/Cases of ARF%	Risk Factors for ARF Post Cardiac Catheterization
1—Clinical and angiographic profile in coronary artery disease: Hospital outcome with emphasis on the very elderly [19] 2010 LILACS	Galon et al.	The study of 1282 patients who underwent 1410 cardiac catheterizations, selected from March 2007 to May 2008 from a database in a general hospital for diagnosis of coronary artery disease (CAD).	1.282/1.6%	Elderly from 75. Presence of chronic disease. Uses of non-steroidal anti-inflammatories (NSAIDs).
2—Influence of the timing of cardiac catheterization and amount of contrast media on acute renal failure after cardiac surgery [20] 2011 PubMed	Sadeghi et al.	1177 patients undergoing different types of surgery after cardiac catheterization. The influence of time interval and amount of contrast in postoperative ARF were evaluated.	1177/53.57%	Elderly. Previous renal disease. High volumes of contrast. Longer procedure time.
3—Acute renal damage after using percutaneous coronariography: related risk factors [21] 2017 LILACS	González AH, Morejón CCDDS, Barbeito CTOT.	A descriptive and analytic study of 37 patients that required percutaneous coronariography was carried out in the Cardiology Center of “Carlos J. Finlay” Clinical-Surgical Hospital in Havana from October 2009 to January 2010.	37/10.8%	Hypertension Diabetes Mellitus Elderly.
4—Acute kidney injury after contrast-enhanced examination among elderly [22] 2014 LILACS	Aoki et al.	Longitudinal cohort study conducted at the Federal University of São Paulo Hospital from March 2011 to March 2013. All hospitalized elderly, of both sexes, aged 60 years and above, who performed the examination, were included (*n* = 93).	93/54%	Elderly. Hypertension. Diabetes mellitus. Heart failure. Use of non-steroidal anti-inflammatory drugs. High volume of contrast used greater than 200 mL.
5—Coronary angioplasty performed with a total volume of three milliliters of contrast [23] 2011 LILACS	Monteiro et al.	A case study of coronary stent in a patient with chronic renal failure and acute coronary syndrome, using 3 mL of contrast, using the injection system (ACIST) Medical Systems, Eden Prairie, United States and intracoronary ultrasound.	1/100%	Elderly. Diabetes mellitus. Previous renal disease.
6—Baseline atrial fibrillation is associated with contrast-induced nephropathy after cardiac catheterization in coronary artery disease: systemic review and meta-analysis [24] 2018 PubMed	Prasitlumkum et al.	Association between atrial fibrillation in patients with coronary arterial disease and contrast nephropathy through a systematic review of the literature and meta-analysis.	Participants without atrial fibrillation 15,661/43.3% Participants with atrial fibrillation 1030/51.7%	Atrial fibrillation. Diabetes mellitus. Elderly. Heart failure. Hypotension. High volume of contrast used.
7—Contrast-induced nephropathy after cardiac catheterization: a prospective study of 180 patients [25] 2012 PubMed	Mghaieth et al.	In this prospective single-center study, 180 consecutive patients who underwent cardiac catheterization were enrolled; all patients were followed up for 3 months.	180/17.2%	Hypotension.
8—Contrast-induced nephropathy in acute coronary syndrome [26] 2011 PubMed	Carnevalini et al.	In a retrospective cohort, we analyzed consecutive patients hospitalized for acute coronary syndrome undergoing urgent percutaneous coronary intervention (PCI) within 72 h from the admission. Contrast-induced nephropathy was defined as a 25% increase of creatinine levels from baseline at 48 h from the PCI. The inclusion period was from January 1, 2004 to June 30, 2010. A total of 125 patients were analyzed	125/10.4%	Elderly High volume of contrast used.
9—Clinically significant contrast induced acute kidney injury after non-emergent cardiac catheterization--risk factors and impact on length of hospital stay [27] 2013 PubMed	Kashif, Khawaja, Yaqub, Hussain.	Case records of patients who underwent coronary angiography with a serum creatinine of 1.5 mg/dL at the time of procedure were evaluated. Clinically significant contrast-induced nephropathy (CSCIN) was defined as either the doubling of serum creatinine from the baseline value within a week following the procedure or the need for emergency hemodialysis after the procedure.	116/17%	Previous kidney disease Heart failure.
10—Trends in contrast volume use and incidence of acute kidney injury in patients undergoing percutaneous coronary intervention: insights from Blue Cross Blue Shield of Michigan Cardiovascular Collaborative (BMC2) [28] 2018 PubMed	Gurm, Seth, Dixon, Kraft, Jensen	The study population for this analysis included all consecutive patients who underwent PCI between January 2010 and December 2016 at 48 hospitals participating in the BMC2 (Blue Cross Blue Shield of Michigan Cardiovascular Consortium). A total of 182,196 patients underwent PCI over the 7 study years across 48 hospitals.	182.196/25%	High volume of contrast used.

Source: Authors’ Research.

**Table 3 ijerph-17-03392-t003:** Prevalence of risk factors in the sample of this study.

Risk Factor	%
Elderly	70%
Presence of chronic disease *	50%
High volume of contrast used	50%
Heart Failure	30%
Previous kidney disease	30%
Uses of non-steroidal anti-inflammatories	20%
Hypotension	20%
Longer procedure time	10%
Atrial fibrillation	10%

* (hypertension and diabetes mellitus).

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
