# Peer review of "Risk Factors for Acute Renal Failure after Cardiac Catheterization Most Cited in the Literature: An Integrative Review"

_ijerph, 2020, doi:10.3390/ijerph17103392_

Round 1
Reviewer 1 Report
The English language in the manuscript (both grammar and choice of words) should be much improved, as it is hard to read.
In general it is unclear how systematic the authors have carried out the review (and as the references to RIL are in Poruguese (?) I am not able to compare). Clearly this manuscript does not show transparently that the study was carried out systematically.
Maior comments:
Page 2 line 65-73: It is unclear why this section is in the manuscript, as treatment is not included in this study.
Page 3 line 112: "Exclusion includes..." sounds like there are aother, unmentioned exclusion cirteria. Are there?
Page 3 line 140: What are "reading criteria"?
page 3 line 140-142: I am surprised, that you only found 2 relevant papers from PubMed, especially as you in you discussion refer to additional papers, I would expect are indexed in PubMed and seem like they should fulfill your criteria.
Page 4 flowchart: The numbers do not add up: Should 577 be 557? And did you only screen 6 papers (it sounds like you screened 563). And did you really carry out quantitative analysis on 6 papers? If so, you should report on the quantitative analysis. The 0 for qualitative analysis also seems to be wrong, maybe this should be 6?
Page 5 tabel: It is unclear how the evidence levels here relate to those in the methods section. They seem arbitrarly and inconistent. The order of papers seems arbitrary.
Page 6 line 159: It is not at all surprising that LILACS and PubMed dominate, when you only scan in those databases. The sentence does not make sense.
Page 6 line 165-172: This section is just repeating part of the table.
Page 6-8: The discussion section does not really use / relate to the results of this study.
Minor comments:
Page 1, line 9: Affiliation 3 is not used.
Page 1, line 26: The last sentence "Repercusssion ..." does not make sense at all.
Page 1, line 38 + 41: It is unclear, what "DCs" and "CO" are.
Page 3 line 143: Do you mean "PRISMA" instead of "prism"?
Page 3 line 145: Why do you write "Belem (PA), Brazil, 2019" herE?
Page 4 line 147: "Source Author's research" is confusing.
Page 4 line 148-150: This is a repetition form further above
Page 6 line 182: Should "contracts " be "contrasts"?
Page 9 line 306: The "Authorship..." sentence seems like misplaced copy-paste.
Page 9 line 308: The quotation makes do not make sense here.
Author Response
all corrections from the two reviewers were made, marked in yellow

Reviewer 2 Report
In this manuscript, authors performed Integrative Literature Review using LILACS and PubMed databases between 2009 and 2019, and showed that elderly, hypertension and diabetes, previous kidney disease, hypotension, heart failure, higher contrast volumes, use of non-steroidal anti-inflammatory drugs, and atrial fibrillation were identified as risk factors for acute renal failure after cardiac catheterization. The subject of study seems to be interesting. However, there are some serious concerns in this study. The reviewer’s comments are described as follows.
1. The most serious concern is the selected health science descriptors. Despite a number of previous publications regarding cardiac catheterization-induced acute renal failure, only few papers were included in this analysis. This study cannot reflect the current status in cardiovascular and nephrology fields. In order to solve this problem, authors should include “acute kidney injury” and “acute renal failure” in addition to “nephropathies” as health science descriptors. Furthermore, “Cardiac catheterism” is uncommon and unacceptable medical term. “Cardiac catheterization” and “Coronary catheterization” should be used as health science descriptors.
2. In Figure 1, authors should describe the reasons why they excluded the 577 articles from this analysis.
3. In this study, authors just listed the references and potential risk factors in Table 1. Such data presentation is woefully inadequate. Authors have to show which risk factors had greater effects on the development of acute renal failure than others, by calculating hazard ratio for each risk factor and presenting forest plots of the data as new figures.
4. Authors searched publications between 2009 and 2019. Since 2009, the definition of acute kidney injury has been established by RIFLE, AKIN, and KDIGO criteria. Based on these criteria, associations between risk factors and stages or severity of acute kidney injury should be evaluated.
Author Response
all corrections of the two reviewers were made, marked in yellow

Round 2
Reviewer 2 Report
In the revised manuscript, authors improved search strategy and added four more references. However, as described in the original comments, authors' quantitative analysis is inadequate. They just showed the prevalence of potential risk factors. The reviewer requested to calculate hazard ratio for each factor and present forest plots of the data as a new figures. Authors have to explain how strongly each potential risk factor was associated with the development of acute renal failure in meta-analysis using appropriate statistical methodology. Evaluation of heterogeneity among the extracted papers is also important. Unless adequate statistical analysis is performed, any conclusions from this study cannot be accepted.
Author Response
Doc WORD anexado
